# Active Pulmonary Tuberculosis in Elderly Patients: A 2016–2019 Retrospective Analysis from an Italian Referral Hospital

**DOI:** 10.3390/antibiotics9080489

**Published:** 2020-08-07

**Authors:** Francesco Di Gennaro, Pietro Vittozzi, Gina Gualano, Maria Musso, Silvia Mosti, Paola Mencarini, Carlo Pareo, Antonino Di Caro, Vincenzo Schininà, Enrico Girardi, Fabrizio Palmieri

**Affiliations:** 1Respiratory Infectious Diseases Unit, National Institute for Infectious Diseases “L. Spallanzani” IRCCS, 00149 Rome, Italy; pietro.vittozzi@inmi.it (P.V.); gina.gualano@inmi.it (G.G.); maria.musso@inmi.it (M.M.); silvia.mosti@inmi.it (S.M.); paola.mencarini@inmi.it (P.M.); carlo.pareo@inmi.it (C.P.); fabrizio.palmieri@inmi.it (F.P.); 2Microbiology and Bio-Repository Unit, National Institute for Infectious Diseases “L. Spallanzani” IRCCS, 00149 Rome, Italy; antonino.dicaro@inmi.it; 3Diagnostic Imaging Unit, National Institute for Infectious Diseases “L. Spallanzani” IRCCS, 00149 Rome, Italy; vincenzo.schinina@inmi.it; 4Clinical Epidemiology Unit, National Institute for Infectious Diseases “L. Spallanzani” IRCCS, 00149 Rome, Italy; enrico.girardi@inmi.it

**Keywords:** tuberculosis (TB), elderly, outcome, adverse events, risk factors

## Abstract

Tuberculosis (TB) in the elderly (>65 years old) has increasingly become a global health problem. It has long been recognized that older people are vulnerable to developing tuberculosis. We retrospectively evaluated data from patients older than 65 years diagnosed with pulmonary TB admitted to the National Institute for Infectious Diseases L. Spallanzani, Rome, Italy, from 1 January 2016 to 31 December 2019. One hundred and six consecutive patients were diagnosed with pulmonary TB and 68% reported at least one comorbidity and 44% at least one of the TB risk-factors. Out of the 26 elderly patients who reported an adverse event, having risk factors for TB (O.R. (Odds Ratios) = 1.45; 95% CI 1.12–3.65) and the presence of cavities on Chest X-rays (O.R. = 1.42; 95% CI 1.08–2.73) resulted in being more likely to be associated with adverse events in elderly patients. Having weight loss (O.R. = 1.31; 95% CI 1.08–1.55) and dyspnea (O.R. = 1.23; 95% CI 1.13–1.41) resulted in being significant predictors of unsuccessful treatment outcome in elderly patients. Older people with TB represent a vulnerable group, with high mortality rate, with a challenging diagnosis. Hospitalizations in tertiary referral hospital with clinical expertise in TB management can be useful to improve the outcome of these fragile patients.

## 1. Introduction

Despite extensive tuberculosis-control efforts of the World Health Organization (WHO) and local health departments, the tuberculosis (TB) epidemic continues to ravage the world, affecting susceptible individuals including the elderly (>65 years old), and representing a global health problem [1,2]. The geriatric population in a high income setting such as in Italy represents a large reservoir of TB infection across all sexual and gender subgroups. Age-related comorbidities (e.g., malnutrition, cancer, chronic renal failure, and diabetes mellitus), together with physiological biological changes may weaken protective barriers, impair microbial clearance mechanisms, and contribute to reducing cellular immune responses to *M. tuberculosis*, thus increasing the risk of TB among this age group [3]. Moreover elderly people are both at especially high risk for reactivation of latent TB, but also susceptible to new TB infection. Tuberculosis diagnosis in the elderly can be challenging; in fact, elderly patients may have an absent or attenuated febrile response with a high frequency of nonspecific clinical manifestations or co-morbidities and less frequently “classical” radiological presentations that can result in delay in diagnosis [4]. Older patients ≥65 years of age have a higher mortality rate compared to patients under the age of 65 [5]. In fact, global data from low incidence countries show that around 80% of the deaths occur in patients over sixty-five years of age [6]. The mortality rates of older patients have been reported as up to 51% [7]. Although these mortality rates have been decreasing recently, the rate remains high [8,9]. Moreover, age represents a risk factor connected to the development of adverse drug reactions due to polypharmacy, pill burden, existing co-morbidities, and a lower efficiency of renal and hepatic drugs clearance. [6,10]

To our knowledge, there are no data in Italy investigating and reporting findings from this vulnerable group. Therefore, we conducted a retrospective study in elderly pulmonary TB patients in order to describe clinical presentation and factors associated with adverse events and outcome.

## 2. Results

Between 1 January 2016 and 31 December 2019, 106 consecutive patients age >65 (median age years 76, IQR 5; male n. 73, 69%) were diagnosed with pulmonary TB at INMI L. Spallanzani Institute in Rome and included in the study.

Table 1 shows the characteristics of participants overall and stratified by age-classes ≤75 or >75 years old. Of the whole sample, 68% (*n* = 73) reported at least one comorbidity, and 44% (*n* = 47) at least one of the TB risk-factors between diabetes, chronic renal failure, malignances, and being under immunosuppressive therapy. Forty patients out of 106 (37%) were sputum smear positive. Notably, 52% (*n* = 55) of the participants referred non-specific initial symptoms, and 17% (*n* = 18) also had an extrapulmonary localization of TB.

Adverse events related to TB drug regimen were reported in 24% (*n* = 26). Ninety-one per cent (*n* = 96) of patients successfully completed the treatment, while 8% (*n* = 9) died due to a TB-related cause and one failed. No differences in distribution of the variables collected between ≤75 or >75 year-old groups emerged (*p*-value > 0.05).

Out of the 26 elderly patients who reported an adverse event, 69% (*n* = 18) showed liver disease. In 16 patients (61%), the event required minimal non-invasive intervention classified as of moderate severity and 81% (*n* = 21) of cases required a suspension of suspect drug and change of treatment. The new therapeutic scheme after suspension included fluoroquinolones in 15 out of 21 patients (71%). Further characteristics of adverse events reported in the sample are shown in Table 2.

The multivariate model on adverse events considered the effects of age, gender, risk factors for TB, comorbidities, fever, cough, dyspnea, cavities on CXR, drug without Z regimen including fluorquinolone, extrapulmonary TB, acid-fast bacilli smear positive, and TB culture. Significant predictors of adverse events are reported in Table 3. Having risk factors for TB (O.R. = 1.45; 95% CI 1.12–3.65) and presence of cavities on CXR (O.R. = 1.42; 95% CI 1.08–2.73) resulted in being more likely to be associated with adverse events in elderly patients.

The multivariate model on unsuccessful outcome considered the effects of age, gender, risk factors for TB, comorbidities, fever, cough, dyspnea, weight loss, cavities on CXR, drug without Z regimen including fluorquinolone, extrapulmonary TB, acid-fast bacilli smear positive, and culture positive. Having weight loss (O.R. = 1.31; 95% CI 1.08–1.55) and dyspnea (O.R. = 1.23; 95% CI 1.13–1.41) resulted in being significant predictors of unsuccessful treatment outcome (death or failure) in elderly patients, as reported in Table 4

## 3. Discussion

Our cohort describes a large group of elderly patients from a tertiary referral hospital for TB in Italy. According to the WHO definitions, in our cohort, 91 out of 106 patients (86%) were bacteriologically confirmed with pulmonary TB.

As highlighted by the WHO, TB in elderly patients represents one of the pilot keys for high income countries to control TB burden [11,12,13]. TB and age are strongly related since underlying illness and the physiological reduction of the immune system of the elderly make this group of people more susceptible to a reactive latent form or to a new infection [14,15].

Previous data showed that co-morbidities such as diabetes, COPD, and cardiac disease, which are prevalent in aging populations, increase the risk of developing active TB disease [16,17]. In this cohort, more than half of the patients had one or more comorbidities, and presented risk factors for developing TB. Despite age and comorbidities, 96 (91%) patients were defined as successfully treated. Unsuccessful outcome was reported in 10 out of 106 patients (9%), where nine patients died.

Unlike other studies, there was no difference in unsuccessful treatment between age <75 years and >75 years in TB patients and the mortality rate was only 8.5% (9/106) in older patients [18]. Our data showed lower mortality rates than reported in other studies worldwide (up to 51%) [19,20] and in European low incidence countries (from 11.2% in UK to 29.3 in Spain) [8,21].

We believe that the low mortality and high treatment success rates in this retrospective study could reflect the accurate diagnosis, coupled with the opportunity to test for drug susceptibility, timely monitoring, and treatment of adverse events, and close follow up after discharge. Moreover, two elements can have contributed: the Italian health system, which provides universal coverage to citizens and residents free of charge, and the setting of the study, a tertiary referral hospital with clinical expertise in the diagnosis and treatment of TB. Furthermore, our experience confirmed, according to the WHO statements, the role of active TB drug-safety monitoring, aDSM, especially in the elderly, to minimize the unsuccessful treatment [22,23].

However, the data findings underline the vulnerability of these patients, the difficulty in achieving therapeutic success compared to other categories, and the need for clinical attention.

Two recent meta-analyses investigated the factors associated with a worse treatment outcome. The authors found that age, sex, alcohol consumption, smoking, sputum smear non-conversion at two months of treatment, and HIV affect the results of TB treatment. From their research, no clinical data influenced TB outcome [24,25].

In our cohort, 52% (*n* = 55) of the participants referred non-specific initial symptoms, and 17% (*n* = 18) also had an extrapulmonary localization of TB. We found that weight loss and dyspnea at diagnosis were negative predictive factors for negative outcome.

Diagnostic difficulties in the elderly are common in many diseases, not solely TB, and the development of new diagnostic tools such as ultrasound and point of care diagnostic can help overcome this important issue [26,27].

Many papers have been published with regard to clinical and radiologic presentation of TB in varying ages. Patients may present a lack of respiratory symptoms and may be unable to expectorate sputum due to weakness. In a comparison between clinical features in the young and old (adults), the classical symptoms of productive cough, night sweats, fever, weight loss, and hemoptysis were much less common in the older age group [28,29].

In our cohort, fever, cough, dyspnea, and weight loss were the most common symptoms. We found that dyspnea and weight loss at the time of diagnosis were predictive factors for negative outcome. Dyspnea and weight loss could have been associated with a more advanced stage of the disease, which has been more affected by diagnostic delay, and with poorer clinical conditions. Many studies demonstrated how advanced stages of illness were associated with unfavorable outcome, underlining the pivot role of early recognition of symptoms and rapid diagnosis and therapy for TB outcome [30].

Radiological findings represent another challenging diagnosis element [31,32]. Imaging could provide suspects for diagnosis and timely treatment for active TB. Consistently with other data in literature, only 32 patients (29%) showed typical radiographs of upper lobe cavitary TB lesions, while another study showed a smaller proportion (16%) of patients with typical radiological manifestations [33].

Adverse events (AEs) represent an important issue in a long therapy like TB. In our cohort, 26 (24%) patients developed AEs (six were severe AE); the most common was liver disfunction (69%).

Elderly people are more likely to develop adverse drug interaction from polypharmacy, pill burden, existing co-morbidities, and age-related physiological changes. Many reasons can explain the high rate of adverse events: active research by clinicians of adverse events during hospitalization or outpatient controls, hepatotoxicity from anti-tuberculous drugs rises significantly with increasing age, visual impairment, poor memory, and reduced mobility may cause poor adherence to the drug regimen [34,35].

From our data, predictors of adverse events for TB in elderly patients were: having risk factors for TB (diabetes, malnutrition, alcohol abuse, immunosuppressive therapy) and cavities on CXR. Additionally, other studies have shown how risk factors to onset TB were independent predictors of adverse drug events. Thus, closely monitored follow-ups are highly recommended in these patients [36,37].

Consistent with the other data in the literature, our study identified Pyrazinamide and Isoniazid as drugs responsible for AEs [38,39,40].

Contrary to other studies, no mortality can be attributed to drugs [41]. Discontinuation of suspected drugs required a change in treatment regimen in 21 patients who developed AEs.

In 26 patients, first line regimen was replaced by fluoroquinolones (FQs) and in 15 patients, FQs were added after suspension due to AEs. No significant AEs was found in patients receiving FQs. International warnings have recommended that it is used cautiously in elderly patients and those with renal impairment due to possible QTc prolongation, increased risk for drug accumulation, and in patients with diabetes or those who are taking hypoglycemic agents due to the risk for severe hypoglycemia [42,43]. In our cohort, 41 patients (39%) were on fluoroquinolones. We found only one cardiac adverse event (arrhythmia). Despite other evidence, in our experience, FQs were well tolerated drugs [44].

We recognize some limitations in our study: the enrolment of patients diagnosed in one institution may limit the extent to which our findings can be generalized. The retrospective nature of the study precluded the consideration of other factors potentially influencing outcome such as lack of socioeconomic characteristics and evaluation of diagnostic delay [25].

## 4. Materials and Methods

### 4.1. Study Design and Patients

This is a retrospective study in elderly Italians diagnosed with pulmonary TB admitted to the National Institute for Infectious Diseases L. Spallanzani, Rome, Italy, from 1 January 2016 to 31 December 2019. The study protocol was approved by L. Spallanzani Institute Ethics Committee (Decision No. 12/2015)

We retrospectively evaluated data from patients older than 65 years old with pulmonary active TB. According to the WHO guidelines, enrolled patients were classified as “active pulmonary TB” if the diagnosis was based: (i) on a positive culture for *M. tuberculosis* from a respiratory sample (sputum or bronchoalveolar lavage) or other biological specimens; (ii) on positive *M. tuberculosis* nucleic amplification test NAT (TRCReady^®^ MTB, Tosoh Bioscience, Japan or Xpert^®^ MTB/RIF, Cepheid, USA) from biological specimens (without culture confirmation); or (iii) on histopathological findings consistent with TB and presence of acid fast bacilli (AFB) in a tissue sample. Moreover, patients were classified as ‘‘clinical TB’’ if the diagnosis was based on clinical and radiologic criteria (having excluded other diseases) including appropriate response to standard anti-TB therapy.

Patients were treated according to the institutional protocol drawn up following the WHO TB guidelines [11]. Initial treatment was provided on an in-hospital basis, until AFB sputum conversion was achieved on three consecutive negative samples collected during one week. Patients received direct observed therapy during hospitalization. After discharge, patients were followed monthly on ambulatory care by trained TB specialists for the full course of treatment. Laboratory tests were repeated almost monthly, or as needed according to clinical condition, until the treatment was completed. Medical counseling was given regarding the possibility of adverse drug reaction occurrence and the importance of reporting any adverse event to their physician as soon as it presented. Patients were also provided with an information sheet about the most common side effects of anti-TB drugs.

### 4.2. Data Collection

The primary data source was the patient chart, from where a set of predefined variables were retrospectively collected by a study physician and included patient demographics; admission and discharge/death date; and clinical variables: symptoms, comorbidities, risk factors for TB, TB diagnosis, *M. tuberculosis* drug resistance, chest x-ray (CXR) findings, TB localization, treatment regimen, adverse events (type, severity, management), and outcomes. As comorbidities, previous TB, COPD/bronchiectasis, cardiopathy, hypothyroidism, dementia, hematological disease, and chronic liver disease were included. Risk factors for TB included diabetes, chronic renal failure, malignances, and being under immunosuppressive therapy.

Severity of adverse events were classified as mild (asymptomatic laboratory findings only; minor signs/symptoms; no medical intervention required), moderate (requiring minimal non-invasive intervention), and severe (significant symptoms requiring hospitalization) [11].

### 4.3. Statistical Analysis

No formal sample size was calculated a priori, and the study included all patients older than 65 years admitted during the study period. Continuous data were expressed as median and interquartile range (IQR), and categorical data as numbers and percentages. X2 test (with Fisher’s correction as required) was used to compare categorical variables. We stratified our cohort into two groups: <75 years of age and older than 75 years in order to verify any differences between these two groups.

Two logistic regression models were implemented considering the dependent variables as: (I) the adverse events; and (II) unsuccessfully treated (died and failed patients) while each of the available factors were considered as independent variables (univariate analysis). In the multivariate analysis, all the factors with a *p*-value < 0.10 at the univariate analyses were included. Multicollinearity among covariates was assessed through the variance inflation factor (VIF), taking a value of two for excluding a covariate. However, no variable was excluded according to the previous criterion.

Odds ratios (ORs) as adjusted odds ratios (Adj–ORs) with their 95% confidence intervals (CIs) were used to measure the association between factors at the baseline (exposure) and treatment failure (outcome).

All analyses were 2-sided and a *p*-value less than 0.05 was considered statistically significant. Statistical analysis was performed using STATA V.13.

## 5. Conclusions

In conclusion, our data suggest that older people with TB represent a vulnerable group with high mortality rate, a challenging diagnosis due to a higher frequency of atypical features, symptoms common with other age-related diseases, and more frequent adverse drug reactions. Dyspnea and weight loss at TB diagnosis are risk factors for treatment failure and describe the phenotype of a patient who needs more medical attention and close monitoring. Having risk factors for TB also represents a predictor for adverse events. In addition, data suggest the pivotal role of aDSM to improve outcome in older patients. Hospitalization in tertiary referral hospital with clinical expertise in diagnosis and treatment of TB can be useful to improve outcome considering the elderly as fragile patients. New treatment regimens and novel diagnostics, aiming to reduce adverse events of treatment in TB, should also all be specifically evaluated in elderly populations to reduce pill burden, and increase outcome and safety. TB management, control program, and global research agenda need to consider these patients as a possible key to control TB, especially in low burden countries.

## Figures and Tables

**Table 1 antibiotics-09-00489-t001:** Characteristics of participants stratified by age-classes ≤75 or >75 years old.

Characteristics	No. of Total Participants	No. of Participants Aged ≤ 75	No. Of Participants Aged > 75	*p*-Value
	106 (100%)	50 (100%)	56 (100%)	
**Sex, N (%)**				
Female	33 (31%)	14 (42%)	19 (58%)	0.5
Male	73 (69%)	36 (49%)	37 (51%)	
Age, median, (IQR)	76 (5)	70.5 (5)	82 (5)	0.4
Comorbidities, n (%)	72 (68%)	32 (64%)	40 (71%)	0.4
Risk Factors for TB, n (%)	47 (44%)	19 (40%)	28 (60%)	0.2
Type of diagnosis (%)				
Culture positive, n (%)	67 (63%)	28(56%)	39 (70%)	0.3
NAT positive, n (%)	24 (23%)	10 (20%)	14 (25%)	0.4
Histological, n (%)	3 (3%)	2(4%)	1 (2%)	0.5
Clinical, n (%)	12 (11%)	7 (14%)	5 (9%)	0.1
Monoresistance, n (%)	3 (3%)	3 (6%)	0 (0%)	Na
**Initial TB symptoms, n (%)**				
fever	29 (27%)	16 (32%)	13 (23%)	0.2
cough	53 (50%)	26 (52%)	27 (48%)	
weight loss	12 (11%)	8 (16%)	4 (7%)	
night sweats	2 (2%)	1 (2%)	1 (2%)	
hemoptysis	9 (8%)	5 (1%)	4 (7%)	
dyspnea	24 (23%)	10 (20%)	14 (25%)	
Non- specific symptoms *	55 (52%)	22 (44%)	33 (59%)	
Cavities on CXR, n (%)	32 (29%)	15 (30%)	17 (28%)	0.8
Concurrent Extrapulmonary TB, n (%)	18(17%)	8 (16%)	10 (18%)	0.8
**Initial Therapeutic Scheme, n (%)**				
R + H + E + Z	76 (72%)	40 (80%)	36 (64%)	0.4
Drug regimen without Z including Amikacin	4 (3%)	0 (%)	4 (7%)	
Drug without Z regimen including fluoroquinolone	26 (24%)	11(22%)	15 (26%)	
Adverse events, n (%)	26 (24%)	15 (30%)	11 (20%)	0.5
Outcomes, n (%)				
successful treatment	96 (91%)	47 (94%)	49 (87%)	0.2
died	9 (8%)	2 (4%)	7 (13%)	
failure	1(1%)	1(2%)	0 (0%)	

* non-specific symptoms: headache, gastrointestinal.

**Table 2 antibiotics-09-00489-t002:** Characteristics of adverse events in the 26 patients who reported them.

Characteristics	Total
	26 (100)
	**N (%)**
**Type of Adverse events, n (%)**	
Hepatitis	18 (69%)
Ocular damage/ decrease in visual acuity	2 (7%)
itching / skin rash	3 (11%)
arrhythmia	1 (4%)
gastrointestinal	3 (11%)
acute renal failure	1 (4%)
More than 1 Adverse events, n (%)	3 (11%)
**Severity of Adverse events, n (%)**	
Mild	4 (15%)
Moderate	16 (61%)
Severe	6 (23%)
**Adverse events management, n (%)**	
temporary suspension of suspect drug and change of drug	18 (69%)
temporary suspension of all treatment	1 (4%)
definitive suspension of suspect drug	2 (8%)
support therapy and no change of treatment	5 (19%)
**Which drugs, n (%)**	
E	1 (4%)
H	3 (11%)
Z	10 (38%)
Mfx	1 (4%)
>1 drugs (of which 8/11 Z + H)	11 (42%)
**2nd therapeutic scheme, n (%)**	
With FQ	15 (71%)
Without FQ	6 (29%)

**Table 3 antibiotics-09-00489-t003:** Predictors of adverse events for active pulmonary tuberculosis in elderly patients.

Characteristics	Univariate Analysis O.R.	Multivariate Analysis Adj-O.R.
Age <75	1.02 (0.98–1.04)	0.94 (0.77–2.11)
Female	0.28 (0.16–0.40)	0.58 (0.27–1.78)
Risk Factors for TB	1.30 (1.18–1.54)	1.45 (1.12–3.65) *
Comorbidities	0.54 (0.10–1.10)	0.49 (0.28–3.08)
Fever	1.21 (0.28–1.23)	1.33 (1.18–1.54)
Cough	0.64 (0.38–0.78)	0.74 (0.50–1.03)
Dyspnea	1.26 (0.85–1. 72)	1.33 (1.17–3.90)
Cavities on CXR	1.20 (0.09–1.48)	1.42 (1.08–2.73) *
Regimen without Z regimen including fluorquinolone	0.25 (0.10–0.40)	0.78 (0.45–1.74)
R + H + E + Z (Standard regimen)	0.79 (0.68–1.21)	1.01 (0.83–3.21)
Extrapulmonary TB	0.57 (0.59–0.83)	0.51 (0.43–2.70)
Acid-fast bacilli smear positive	1.06 (0.25–1.09)	-
Tb culture positive	1.21 (0.89–1.73)	-

statistically significant **p* < 0.05.

**Table 4 antibiotics-09-00489-t004:** Predictors of unsuccessful treatment outcome in elderly TB patients.

Characteristics	Univariate Analysis O.R	Multivariate Analysis Adj-O.R
Age <75	0.44 (0.29–2.08)	0.55 (0.26–1.06)
Female	0.68 (0.41–0.74)	0.63 (0.43–1.61)
Risk Factors for TB	1.18 (0.81–6.47)	1.36 (0.80–2.90)
Comorbidities	0.90 (0.69–1.21)	1.19 (0.10–1.40)
Adverse events	1.03 (0.88–1.30)	1.18 (0.71–1.25)
Fever	1.01 (0.94–1.95)	1.40 (0.86–1.61)
Cough	0.87 (0.36–1.71)	0.73 (0.50–1.91)
Weight loss	1.28 (1.10–1.44)	1.31 (1.08–1.55) *
Dyspnea	1.18 (1.03–1.74)	1.23 (1.13–1.41) *
Cavities on CXR	1.20 (1.03–1.27)	1.48 (0.54–1.75)
Drug regimen without Z including Fluoroquinolone	1.30 (0.99–2.77)	-
R + H + E + Z	0.88 (0.76–1.09)	1.08 (0.78–1.71)
Extrapulmonary TB	0.65 (0.28–0.85)	1.54 (0.34–3.92)
Acid-fast bacilli smear positive	0.51 (0.33–2.61)	-
TB culture positive	1.45 (0.38–1.77)	0.65 (0.28–2.68)

* statistically significant.

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
