# Peer review of "Active Pulmonary Tuberculosis in Elderly Patients: A 2016–2019 Retrospective Analysis from an Italian Referral Hospital"

_antibiotics, 2020, doi:10.3390/antibiotics9080489_

Round 1
Reviewer 1 Report
The manuscript “Active pulmonary Tuberculosis in elderly patients: a 2016-2019 retrospective analysis from an Italian reference hospital” reports an interesting data on tuberculosis in the elderly. This data would help medical practitioners to understand the complications and comorbidities in elderly people. The results are interesting; however, discussion is shallow. Discussion part covers more general statement than scientific interpretation of data and its reasonable explanation.
For example; Line 108-110, “Although several studies have described the differences between older and younger TB patients, there were some discordant findings on clinical presentation, radiological findings, and treatment outcomes.” Authors must support this statement with suitable references, and include a brief paragraph on ‘differences between older and younger TB patients’ based on their findings supported with reported ones for readers understanding.
Discussion part needs to be revised based on scientific interpretations of findings, and also its relation with already existing data related to complications and comorbidities in elderly people suffering from TB.
Conclusion should also be improved.
Author Response
To Reviewer 1,
We have appreciated the positive feedback and your suggestions to our manuscript “Active pulmonary Tuberculosis in elderly patients: a 2016-2019 retrospective analysis from an Italian referral hospital”. We have answered to all the proposed revisions and we believe that it has remarkably contributed to improve the manuscript.
We have highlighted the modifications made in the revised manuscript using the "track changes" feature.
Also, a native English speaker has been engaged to provide an extended English editing in order to improve the fluency and the readability of the manuscript.
Please find a point-by-point response to the referees’ comments below.
Best regards
Dr. Francesco Di Gennaro
Point-by-point reply
The manuscript “Active pulmonary Tuberculosis in elderly patients: a 2016-2019 retrospective analysis from an Italian reference hospital” reports an interesting data on tuberculosis in the elderly. This data would help medical practitioners to understand the complications and comorbidities in elderly people. The results are interesting; however, discussion is shallow.
1.Discussion part covers more general statement than scientific interpretation of data and its reasonable explanation.
For example; Line 108-110, “Although several studies have described the differences between older and younger TB patients, there were some discordant findings on clinical presentation, radiological findings, and treatment outcomes.” Authors must support this statement with suitable references, and include a brief paragraph on ‘differences between older and younger TB patients’ based on their findings supported with reported ones for readers understanding.
Discussion part needs to be revised based on scientific interpretations of findings, and also its relation with already existing data related to complications and comorbidities in elderly people suffering from TB.
R1. We, following your suggestion, completely reformulated our discussion as following:
“Our cohort describes a large group of elderly patients from a tertiary referral hospital for TB in Italy. According to WHO definitions, in our cohort 91 out of 106 patients (86%) were bacteriologically confirmed pulmonary TB.
As highlighted by WHO, TB in elderly patients represents one of the pilot keys for high income countries to control TB burden [12-13]. TB and age are strongly related since underlying illness and the physiological reduction of immune system of the elderly make this people more susceptible to reactive latent form or to a new infection [14-15].
Previous data showed that co-morbidities such as diabetes, COPD and cardiac disease, prevalent in aging populations, increase the risk of developing active TB disease [16-17] . In this cohort more than half of the patients had one or more comorbidities, and presented risk factors for developing TB. Despite age and comorbidities, 96 (91%) patients were defined successfully treated. Unsuccessful outcome was reported in 10 out of 106 patients (9%), with nine patients died.
Differently from other studies, there was no difference in unsuccessful treatment between age <75 ys and > 75 ys in TB patients and the mortality rate was only 8.5% (9/106) in older patients [18]. Our data show lower mortality rates than reported in other studies worldwide (up to 51%) [19-20] and in European low incidence countries (from 11.2% in UK to 29.3 in Spain) [8, 21]
We believe that the low mortality and high treatment success rates in this retrospective study, could reflect the accurate diagnosis, coupled with the opportunity to test for drug susceptibility, timely monitoring and treatment of adverse events, and close follow up after discharge. Moreover two elements can have contributed: the Italian health system provides universal coverage to citizens and residents, free of charge, and the setting of the study, a tertiary referral hospital with clinical expertise in diagnosis and treatment of TB. Furthermore, our experience confirmed, according to WHO statements, the role of active TB drug-safety monitoring - aDSM, especially in the elderly, to minimize the unsuccessful treatment. [22-23]
Howewer data findings underline the vulnerability of these patients, the difficulty in achieving therapeutic success compared to other categories and need for clinical attention.
Two recent meta-analyses investigated the factors associated with a worse. treatment outcome. Authors find that age, sex, alcohol consumption, smoking, sputum smear non-conversion at two months of treatment and HIV affect the results of TB treatment. From their research no clinical data influence TB outcome [24-25].
In our cohort 52% (n. 55) of the participants referred non-specific initial symptoms, and 17% (n.18) had also an extrapulmonary localization of TB. We found that weight loss and dyspnea at diagnosis were negative predictive factors for negative outcome.
Diagnostic difficulties in the elderly are common in many diseases, not solely TB, and the development of new diagnostic tools, such as ultrasound and point of care diagnostic, can help overcome this important issue [26-27].
Many papers have been published with regards to clinical and radiologic presentation of TB in varying ages. Patients may present a lack of respiratory symptoms and may be unable to expectorate sputum due to weakness. In a comparison between clinical features in the young and old (adults), the classical symptoms of productive cough, night sweats, fever, weight loss and haemoptysis were much less common in the older age group. [28-29]
In our cohort fever, cough dyspnea and weight loss are the most common symptoms. We found that dyspnea and weight loss at the time of diagnosis were predictive factors for negative outcome. Dyspnea and weight loss could have been associated with a more advanced stage of the disease, which has been more affected by diagnostic delay, and with poorer clinical conditions. Many studies demonstrated how advanced stages of illness were associated with unfavorable outcome, underlining the pivot role of early recognition of symptoms and rapid diagnosis and therapy for TB outcome [30].
.
Radiological findings represent another challenging diagnosis element [31-32]. Imaging could provide suspect for diagnosis and timely treatment for active TB. Consistently with other data in literature, only 32 patients (29%) showed the typical radiographs upper lobe cavitary TB lesions, while other study showed a smaller proportion (16%) of patients with typical radiological manifestations [33].
Adverse events (AEs) represent an important issue in a long therapy like TB. In our cohort 26 (24%) patients developed AEs (6 were severe AE) ; the most common was liver disfunction (69%).
Elderly people are more likely to develop adverse drug interaction from polypharmacy, pill burden, existing co-morbidities and age-related physiological changes. Many reasons can explain the high rate of adverse events: active research by clinicians of adverse events during hospitalization or outpatient controls, hepatotoxicity from anti-tuberculous drugs rises significantly with increasing age, visual impairment, poor memory and reduced mobility may cause poor adherence to the drug regimen. [34-35].
From our data, predictors of adverse events for TB in elderly patients were: having risk factors for TB (diabetes, malnutrition, alcohol abuse, immunosuppressive therapy) and cavities on CXR. Also, other studies show how risk factors to onset TB were independent predictors of adverse drug events. Thus, closely monitoring are highly recommended in these patients. [36-37].
Consistently with other data of literature, our study identified Pyrazinamide and Isoniazid as drugs responsible of AEs [38-40].
Contrary to other studies, no mortality can be attributed to drugs [41]. Discontinuation of suspected drug required change in treatment regimen in 21 patients who developed AEs.
In 26 patients first line regimen was replaced by fluoroquinolones (FQs) and in 15 patients FQs were added after suspension due to AEs . No significant AEs was found in patients receiving FQs.International warnings recommended that it be used cautiously in elderly patients and those with renal impairment owing to possible QTc prolongation, increased risk for drug accumulation, and in patients with diabetes or those who are taking hypoglycemic agents owing to the risk for severe hypoglycemia [42-43]. In our cohort 41 patients (39%) were on fluoroquinolones . We found only one cardiac adverse event (arrhythmia) Despite other evidence, in our experience FQs were well tolerated drugs [44].
We recognize some limitations in our study: the enrolment of patients diagnosed in one institution may limit the extent to which our findings can be generalized. The retrospective nature of the study precluded the consideration of other factors potentially influencing outcome, such as lacks of socioeconomic characteristics and evaluation of diagnostic delay [25].
- Conclusion should also be improved.
R2. In conclusion, our data suggest that older people with TB represent a vulnerable group, with high mortality rate, a challenging diagnosis due to a higher frequency of atypical features, symptoms common with other age-related diseases and more frequent adverse drug reactions. Dyspnea and weight loss at TB diagnosis are risk factors for treatment failure and describe the phenotype of a patient who needs more medical attention and close monitoring. Having risk factors for TB also represents a predictor for adverse events. In addition, data suggest the pivotal role of aDSM to improve outcome in older patients. Hospitalization in tertiary referral hospital with clinical expertise in diagnosis and treatment of TB can be useful to improve outcome considering the elderly as fragile patients. New treatment regimens and novel diagnostics, aiming to reduce adverse events of treatment in TB, should all be specifically evaluated also in elderly populations, to reduce pills burden, increase outcome and safety. Tb management, control program and global research agenda need to consider these patients as a possible key to control TB, especially in low burden countries

Reviewer 2 Report
This is a study that retrospectively describes a TB from an Italian Reference Hospital. The study is helpful for readers and hospital personnel. However, some improvement can be done to further assist the readers to understand.
I. The discussion is too random. Please follow the structure of the result presentation and then lay out the discussion. Also, the discussion can be shortened.
II. The English is difficult to understand. I only listed a few below and please consult an English native speaker to edit.
95-96: please combine this sentence with the next paragraph (single sentence can not stand as a paragraph)
107-108: "Furthermore, some features became this people more complicated to make diagnosis and frequent diagnostic delay has been reported [14]." What does "this people" mean??
111-112: please combine this sentence with the previous paragraph
113-114: In our cohort, fever, cough and weight loss are the most common symptoms and these can be confused as pneumonia or cancer [17-18].
128-129: In fact, both studies from Spain (29.3%) than (than or and?) UK (11.2%) shown higher mortality rate compared with our data [25-26].
139-141: the Italian health system which(delete) provides universal coverage to citizens and residents, free of charge, and the setting of the study, a tertiary
referral hospital with clinical expertise in diagnosis and treatment of TB.
Author Response
To Reviewer 2,
We have really appreciated your feedback and suggestions to our manuscript “Active pulmonary Tuberculosis in elderly patients: a 2016-2019 retrospective analysis from an Italian referral hospital ”. We have answered to all the proposed revisions and we believe that it has remarkably contributed to improve the manuscript.
We have highlighted the modifications made in the revised manuscript using the "track changes" feature.
Also, a native English speaker has been engaged to provide an extended English editing in order to improve the fluency and the readability of the manuscript.
Please find a point-by-point response to the referees’ comments below.
Best regards
Dr. Francesco Di Gennaro
Point-by-point reply
Reviewer 2
Comments and Suggestions for Authors
This is a study that retrospectively describes a TB from an Italian Reference Hospital. The study is helpful for readers and hospital personnel. However, some improvement can be done to further assist the readers to understand.
- The discussion is too random. Please follow the structure of the result presentation and then lay out the discussion. Also, the discussion can be shortened.
- The English is difficult to understand. I only listed a few below and please consult an English native speaker to edit.
95-96: please combine this sentence with the next paragraph (single sentence can not stand as a paragraph)
107-108: "Furthermore, some features became this people more complicated to make diagnosis and frequent diagnostic delay has been reported [14]." What does "this people" mean??
111-112: please combine this sentence with the previous paragraph
113-114: In our cohort, fever, cough and weight loss are the most common symptoms and these can be confused as pneumonia or cancer [17-18].
128-129: In fact, both studies from Spain (29.3%) than (than or and?) UK (11.2%) shown higher mortality rate compared with our data [25-26].
139-141: the Italian health system which(delete) provides universal coverage to citizens and residents, free of charge, and the setting of the study, a tertiary
referral hospital with clinical expertise in diagnosis and treatment of TB.
R Following your suggestion, we re-wrote the discussion as below:
“According to WHO definitions, in our cohort 91 out of 106 patients (86%) were bacteriologically confirmed pulmonary TB. As highlighted by WHO, TB in elderly patients represents one of the pilot keys for high income countries to control TB burden [12-13]. TB and age are strongly related since underlying illness and the physiological reduction of immune system of the elderly make this people more susceptible to reactive latent form or to a new infection [14-15].
Previous data showed that co-morbidities such as diabetes, COPD and cardiac disease, prevalent in aging populations, increase the risk of developing active TB disease [16-17] . In this cohort more than half of the patients had one or more comorbidities, and presented risk factors for developing TB. Despite age and comorbidities, 96 (91%) patients were defined successfully treated. Unsuccessful outcome was reported in 10 out of 106 patients (9%), with nine patients died.
Differently from other studies, there was no difference in unsuccessful treatment between age <75 ys and > 75 ys in TB patients and the mortality rate was only 8.5% (9/106) in older patients [18]. Our data show lower mortality rates than reported in other studies worldwide (up to 51%) [19-20] and in European low incidence countries (from 11.2% in UK to 29.3 in Spain) [8, 21]
We believe that the low mortality and high treatment success rates in this retrospective study, could reflect the accurate diagnosis, coupled with the opportunity to test for drug susceptibility, timely monitoring and treatment of adverse events, and close follow up after discharge. Moreover two elements can have contributed: the Italian health system provides universal coverage to citizens and residents, free of charge, and the setting of the study, a tertiary referral hospital with clinical expertise in diagnosis and treatment of TB. Furthermore, our experience confirmed, according to WHO statements, the role of active TB drug-safety monitoring - aDSM, especially in the elderly, to minimize the unsuccessful treatment. [22-23]
Howewer data findings underline the vulnerability of these patients, the difficulty in achieving therapeutic success compared to other categories and need for clinical attention.
Two recent meta-analyses investigated the factors associated with a worse. treatment outcome. Authors find that age, sex, alcohol consumption, smoking, sputum smear non-conversion at two months of treatment and HIV affect the results of TB treatment. From their research no clinical data influence TB outcome [24-25].
In our cohort 52% (n. 55) of the participants referred non-specific initial symptoms, and 17% (n.18) had also an extrapulmonary localization of TB. We found that weight loss and dyspnea at diagnosis were negative predictive factors for negative outcome.
Diagnostic difficulties in the elderly are common in many diseases, not solely TB, and the development of new diagnostic tools, such as ultrasound and point of care diagnostic, can help overcome this important issue [26-27].
Many papers have been published with regards to clinical and radiologic presentation of TB in varying ages. Patients may present a lack of respiratory symptoms and may be unable to expectorate sputum due to weakness. In a comparison between clinical features in the young and old (adults), the classical symptoms of productive cough, night sweats, fever, weight loss and haemoptysis were much less common in the older age group. [28-29]
In our cohort fever, cough dyspnea and weight loss are the most common symptoms. We found that dyspnea and weight loss at the time of diagnosis were predictive factors for negative outcome. Dyspnea and weight loss could have been associated with a more advanced stage of the disease, which has been more affected by diagnostic delay, and with poorer clinical conditions. Many studies demonstrated how advanced stages of illness were associated with unfavorable outcome, underlining the pivot role of early recognition of symptoms and rapid diagnosis and therapy for TB outcome [30].
Radiological findings represent another challenging diagnosis element [31-32]. Imaging could provide suspect for diagnosis and timely treatment for active TB. Consistently with other data in literature, only 32 patients (29%) showed the typical radiographs upper lobe cavitary TB lesions, while other study showed a smaller proportion (16%) of patients with typical radiological manifestations [33].
Adverse events (AEs) represent an important issue in a long therapy like TB. In our cohort 26 (24%) patients developed AEs (6 were severe AE) ; the most common was liver disfunction (69%).
Elderly people are more likely to develop adverse drug interaction from polypharmacy, pill burden, existing co-morbidities and age-related physiological changes. Many reasons can explain the high rate of adverse events: active research by clinicians of adverse events during hospitalization or outpatient controls, hepatotoxicity from anti-tuberculous drugs rises significantly with increasing age, visual impairment, poor memory and reduced mobility may cause poor adherence to the drug regimen. [34-35].
From our data, predictors of adverse events for TB in elderly patients were: having risk factors for TB (diabetes, malnutrition, alcohol abuse, immunosuppressive therapy) and cavities on CXR. Also, other studies show how risk factors to onset TB were independent predictors of adverse drug events. Thus, closely monitoring are highly recommended in these patients. [36-37].
Consistently with other data of literature, our study identified Pyrazinamide and Isoniazid as drugs responsible of AEs [38-40].
Contrary to other studies, no mortality can be attributed to drugs [41]. Discontinuation of suspected drug required change in treatment regimen in 21 patients who developed AEs.
In 26 patients first line regimen was replaced by fluoroquinolones (FQs) and in 15 patients FQs were added after suspension due to AEs . No significant AEs was found in patients receiving FQs.International warnings recommended that it be used cautiously in elderly patients and those with renal impairment owing to possible QTc prolongation, increased risk for drug accumulation, and in patients with diabetes or those who are taking hypoglycemic agents owing to the risk for severe hypoglycemia [42-43]. In our cohort 41 patients (39%) were on fluoroquinolones . We found only one cardiac adverse event (arrhythmia) Despite other evidence, in our experience FQs were well tolerated drugs [44].
We recognize some limitations in our study: the enrolment of patients diagnosed in one institution may limit the extent to which our findings can be generalized. The retrospective nature of the study precluded the consideration of other factors potentially influencing outcome, such as lacks of socioeconomic characteristics and evaluation of diagnostic delay [25]”.
- As for the conclusion we improved as follow:
“In conclusion, our data suggest that older people with TB represent a vulnerable group, with high mortality rate, a challenging diagnosis due to a higher frequency of atypical features, symptoms common with other age-related diseases and more frequent adverse drug reactions. Dyspnea and weight loss at TB diagnosis are risk factors for treatment failure and describe the phenotype of a patient who needs more medical attention and close monitoring. Having risk factors for TB also represents a predictor for adverse events. In addition, data suggest the pivotal role of aDSM to improve outcome in older patients. Hospitalization in tertiary referral hospital with clinical expertise in diagnosis and treatment of TB can be useful to improve outcome considering the elderly as fragile patients. New treatment regimens and novel diagnostics, aiming to reduce adverse events of treatment in TB, should all be specifically evaluated also in elderly populations, to reduce pills burden, increase outcome and safety. Tb management, control program and global research agenda need to consider these patients as a possible key to control TB, especially in low burden countries”
- A native English speaker has been engaged to provide an extended English editing in order to improve the fluency and the readability of the manuscript.
95-96: please combine this sentence with the next paragraph (single sentence can not stand as a paragraph)
Thank you for your comments. We united the sentence in a single paragraph.
107-108: "Furthermore, some features became this people more complicated to make diagnosis and frequent diagnostic delay has been reported [14]." What does "this people" mean??
Thank you for your suggestions. We change the word “this people” with “elderly TB patients”
111-112: please combine this sentence with the previous paragraph
Thank you. We modified according to your suggestion.
128-129: In fact, both studies from Spain (29.3%) than (than or and?) UK (11.2%) shown higher mortality rate compared with our data [25-26].
Thank you. We modified as following. In fact, both studies from Spain (29.3%) and UK (11.2%) shown higher mortality rate compared with our data [25-26].
139-141: the Italian health system which(delete) provides universal coverage to citizens and residents, free of charge, and the setting of the study, a tertiary referral hospital with clinical expertise in diagnosis and treatment of TB.
Thank you. We modified according to your suggestion.
The Italian health system provides universal coverage to citizens and residents, free of charge, and the setting of the study, a tertiary referral hospital with clinical expertise in diagnosis and treatment of TB.

Round 2
Reviewer 1 Report
The manuscript is revised well. The authors replied reviewers' comments satisfactorily, therefore this version of manuscript is suggested for acceptance.